# Southern Hemisphere tree-rings as proxies to reconstruct Southern Ocean upwelling

Christian Lewis[1], Rachel Corran[1], Sara E. Mikaloff-Fletcher[3], Erik Behrens[3], Rowena Moss[3], Gordon Brailsford[3], Andrew Lorrey[3], Margaret Norris[1], Jocelyn Turnbull[1,2]

[1]GNS Science, Rafter Radiocarbon Laboratory, Lower Hutt, New Zealand
[2]CIRES, University of Colorado at Boulder, Boulder, Colorado, USA
[3]National Institute of Water and Atmospheric Research (NIWA), Wellington, New Zealand

*Correspondence to*: Dr. Christian Lewis (c.lewis@gns.cri.nz)

**Abstract.** The Southern Ocean plays a key role in regulating global climate and acting as a carbon sink. This region, defined as south of 35°S, is accountable for 40% of all oceanic anthropogenic $CO_2$ uptake, and 75% of ocean heat uptake between 1861 and 2005. However, the strength of the Southern Ocean sink (air-sea $CO_2$ flux) is variable - weakening in the 1990s and strengthening again in the 2000s. Typical methods of constraining the flux must grapple with two opposing forces: outgassing of natural $CO_2$ and uptake of anthropogenic $CO_2$. Reconstructions of atmospheric radiocarbon ($\Delta^{14}CO_2$) from Southern Hemisphere tree-rings may be a viable method of observing the one-way outgassing flux of natural $CO_2$, driven by Southern Ocean upwelling. Here we present 280 tree-ring $\Delta^{14}C$ measurements from 13 sites in Chile and New Zealand from the 1980s to 2017. These measurements dramatically expand the dataset of Southern Hemisphere atmospheric $\Delta^{14}CO_2$ records. We use these records to analyse latitudinal gradients in reconstructed atmospheric $\Delta^{14}CO_2$ across the Southern Ocean. Tree-rings from New Zealand's Campbell Island (52.5S, 169.1E) show $\Delta^{14}CO_2$ was on average 3.1±3.3‰ lower than atmospheric background, driving a latitudinal gradient among New Zealand sites between 41.1°S and 52.5°S, whereas samples from similar latitudes in Chile do not exhibit such a strong gradient. We demonstrate that the gradient is driven by the combination of $CO_2$ outgassing from the Antarctic Southern Zone (ASZ) and atmospheric transport to the sampling sites.

## 1 Introduction

The Southern Ocean (here defined as the ocean region south of 35°S) plays a critical role in Earth's climate and the balance of carbon between the ocean and atmosphere (Anderson et al., 2009; Gruber et al., 2009; Hauck et al., 2023). The region accounts for 40% of all oceanic anthropogenic $CO_2$ uptake, and 75% of ocean heat uptake from 1861 to 2005 (Frölicher et al., 2015; Gruber et al., 2019; Mikaloff Fletcher et al., 2006).

The Pacific, Atlantic, and Indian Oceans all transport carbon-rich deep-waters southward, which upwell in the Southern Ocean, reaching the surface south of the Antarctic convergence (Talley, 2013) and serve as an important control on pre-industrial atmospheric $CO_2$ (Marinov et al., 2006). As atmospheric $CO_2$ mole fractions have increased through the industrial era, these waters are now a net sink for $CO_2$ (Gruber et al., 2019).

Southern Ocean carbon exchange is a balance between two opposing forces: release of natural carbon to the atmosphere and uptake of anthropogenic $CO_2$ into the ocean (Gruber et al., 2019). The balance between these processes determines the net Southern Ocean carbon flux (Gruber et al., 2009). Observational methods have used atmospheric $CO_2$ mole fraction or oceanic $CO_2$ partial pressures (pCO2) to infer the net flux rate, typically in combination with atmospheric and/or ocean modelling. These observational methods of measuring Southern Ocean air-sea $CO_2$ flux must grapple with estimating a small difference between these two opposing forces. Additional challenges are posed by temporal and spatial variability of the air-sea $CO_2$ flux (DeVries et al., 2017; Fay et al., 2014; Gruber et al., 2019, 2023; Landschutzer et al., 2015; Takahashi et al., 2012) and seasonal bias toward sampling in spring and summer (Gray et al., 2018; Gruber et al., 2019; Sallée et al., 2010). These factors have led to a wide range of Southern Ocean sink-strength estimates from various methods including ground-based atmospheric measurement, ship-based hydrography, biogeochemical floats, aircraft based flux-

measurements, biogeochemical models, and inverse modelling (DeVries, 2014; Fong and Dickson, 2019; Gray et al., 2018; Gruber et al., 2019; Le Quéré et al., 2007; Long et al., 2021; Mikaloff Fletcher et al., 2006; Nevison et al., 2016; Sarmiento et al., 2023).

Radiocarbon analysis is a powerful tool for informing the controls on atmospheric $CO_2$, and may provide a new lens through which to observe Southern Ocean flux (Levin et al., 2010; Graven et al., 2012). From the 1950s until the 1980s, the radiocarbon "bomb-spike" caused isofluxes from the biosphere, ocean, and atmosphere to be in severe disequilibrium (Randerson et al., 2002; Turnbull et al., 2017; Levin et al., 2010). However, after decades of equilibration, since the 1980s the bomb spike is no longer the dominant driver of atmospheric radiocarbon-in-$CO_2$ ($\Delta^{14}CO_2$) variability, and the Northern Hemisphere $\Delta^{14}CO_2$ spatial patterns are primarily driven by fossil fuel emissions, while the Southern Hemisphere signature is driven by ocean exchange (Levin et al., 2010; Naegler & Levin, 2009; Randerson et al., 2002; Turnbull et al., 2009).

The ocean exchange that drives atmospheric $\Delta^{14}CO_2$ is the same that determines net Southern Ocean sink strength: outgassing of natural $CO_2$ versus anthropogenic $CO_2$ uptake. The waters transported to the Southern Ocean surface via the global thermohaline circulation have distinct properties – they are oxygen poor, macronutrient and carbon rich, and old (Gray et al., 2018; Le Quéré et al., 2007; Lovenduski et al., 2008; McNichol et al., 2022; Talley, 2013). Outgassing of $CO_2$ from these old waters imparts a distinctly low radiocarbon signature ($\Delta^{14}C$) on the atmosphere, in particular in the Indo-Pacific Sector (Prend et al., 2022). As the atmosphere has been saturated with elevated levels of bomb-$^{14}C$, outgassing is measurable as a one-way flux via depletion of atmospheric $^{14}C$. Long-term high-precision $\Delta^{14}CO_2$ records have detected such upwelling signals (Graven et al., 2012; Levin et al., 2010). However, observation sites are limited, and many records are short or include gaps (Graven et al., 2007; Hua et al., 2021).

Here we propose that Southern Hemisphere tree-ring reconstructions of atmospheric $\Delta^{14}CO_2$ may be used to fill in these gaps and expand the available dataset. Tree-rings are considered the "gold-standard" for recording changes in atmospheric $^{14}C$ over long time scales (Southon et al., 2016). Tree rings are used for calibration to calendar dates via the "crossdating" of rings (Leavitt and Bannister, 2009; Reimer et al., 2009, 2013; Southon et al., 2016; Stuiver and Quay, 1981) and can be used for recent carbon cycle studies via the use of the bomb-spike (Ancapichun et al., 2021; Hua et al., 2013, 2021; Turnbull et al., 2017). While a powerful method, users must beware of pitfalls (Leavitt and Bannister, 2009; Southon et al., 2016) including potential interlaboratory offsets, and missing or false rings (Hua et al., 2021; Turnbull et al., 2017), growing season uncertainty or incorporation of carbon from previous or following growth years. We address these concerns in our methodology.

In this paper, we present a time-series of 280 tree-ring $\Delta^{14}C$ measurements from 13 sites in Chile and New Zealand from 1980-2017. We compare these new tree-ring $\Delta^{14}C$ measurements, and two tree-ring records from Turnbull et al., (2017), to a smoothed long-term record of Southern Hemisphere atmospheric $\Delta^{14}C$ from Baring Head and Cape Grim (Levin et al., 2010; Turnbull et al., 2017), calculating $\Delta\Delta^{14}CO_2$. We find that tree-ring $\Delta\Delta^{14}CO_2$ from New Zealand sites have a strong relationship with latitude (lower $\Delta\Delta^{14}CO_2$ found further south). Atmospheric back-trajectory modelling is used to estimate origins of air masses arriving at each tree-ring site and examined in context. We link air-mass origins with GLODAP surface ocean DIC $\Delta^{14}C$ measurements and polar frontal zone spatial extents. We find that low $\Delta\Delta^{14}CO_2$ at our southernmost New Zealand site (Campbell Island) is driven by air originating in the Indian Sector of the Antarctic Southern Zone which exhibits near year-round outgassing (Gray et al., 2018) with lower surface ocean DIC $\Delta^{14}C$ than the Pacific Sector. This work focuses on establishing the method of using Southern Hemisphere tree-ring $\Delta^{14}CO_2$ measurements as a proxy for Southern Ocean upwelling, and does not focus on temporal trends, which will be addressed in a following work.

## 2 Methods

### 2.1 Tree-ring sampling, measurement and ring count validation

Tree-ring sampling in New Zealand and Chile was conducted during field campaigns in 2016 and 2017. The west coasts of Chile, New Zealand and New Zealand's Subantarctic islands were selected because of the predominant westerly air-flow which has ocean-influence and minimal terrestrial biosphere or anthropogenic influence (Fig 2 a/b, Table 1).

| Country | Site | Latitude (°S) | Longitude (°W) | Species | Ring Code |
|---|---|---|---|---|---|
| Chile | Bahia San Pedro | 40.9 | 73.9 | Podocarpus nubigenus (mañio macho) | BSP-T1-C2 |
| Chile | Bahia San Pedro | 40.9 | 73.9 | Nothofagus betuloides (coigue) | BSP-T2-C1 |
| Chile | Tortel River | 47.8 | 73.6 | Nothofagus betuloides (coigue) | TOR-T6-C1 |
| Chile | Tortel Island | 47.8 | 73.6 | Nothofagus betuloides (coigue) | TOR-T4-C1 |
| Chile | Seno Skyring | 52.5 | 72.1 | Nothofagus betuloides (coigue) | SKY-T3-C1/SKY-T4-C2 |
| Chile | Monte Tarn | 53.7 | 71 | Pilferodendrum wiferum (male spruce) | TAR-T3-C1 |
| Chile | Monte Tarn | 53.7 | 71 | Nothofagus betuloides (coigue) | TAR-T6-C2 |
| Chile | Puerto Navarino, Isla Navarino | 54.9 | 68.3 | Nothofagus betuloides (coigue) | PNV-T1-C1 |
| Chile | Baja Rosales, Island Navarino | 54.9 | 67.4 | Nothofagus betuloides (coigue) | ROS-T1-C1/ROS-T4-C1 |
| Country | Site | Latitude (°S) | Longitude (°E) | Species | Ring Code |
| New Zealand | Baring Head | 41.1 | 174.1 | Pinus radiata | BHD-T1-C1/BHD-T1-C3* |
| New Zealand | Eastbourne | 41.3 | 174.1 | Agathis australis (kauri) | NIK19-T1-C2/NIK23-T1-C2* |
| New Zealand | Haast Beach | 43.9 | 169.0 | Pinus radiata | HAB-T1-C1/HAB-T1-C2 |
| New Zealand | Oreti Beach | 46.4 | 168.2 | Pinus radiata | ORT-T2-C1/ORT-T2-C2 |
| New Zealand | Campbell Island | 52.5 | 169.2 | Picea sitchensus (sitka spruce) | WLT-T2-C2/WLT-T3-C3 |

**Table 1. Tabulation of site names, latitude & longitudes, species, and ring-codes of trees that were selected and cores that were used in the final work. Asterisk indicates that tree cores from Baring Head and Eastbourne sites were previously published in (Turnbull et al., 2017)**

Traditional dendrochronology requires multiple trees within a stand of forest for reliable chronologies from annual growth rings. In this work, trees within stands were deliberately avoided in exchange for isolated specimens, often a few meters from the coast or on clifftops directly sampling ocean air, and avoiding the potential for reassimilation of respired $CO_2$ within the canopy. Trees were selected based on location close to the coast with minimal local land influences, feasibility of sampling (clear access to trunk and stable ground), clear and consistent rings and consistency of species across multiple sites.

In short, cores were taken using a 4.3mm increment corer (Haglöf Sweden), removed from the corer and stored in straws to avoid damage during moving or storage. Tree cores were brought back to the Rafter Radiocarbon Laboratory (RRL) for sample workup and measurement. Cores were mounted onto aluminium foil coated cardboard with rubber bands, avoiding the glue traditionally used in dendrochronology. Only cores with clearly defined rings and unambiguous ring counts were considered for further analysis. These cores were sliced into single rings. All cores were sliced by a single person to ensure that any human bias in determining ring boundaries is consistent across all cores sampled in this study.

Sampled rings were then sliced longitudinally into thin "matchsticks" using a scalpel blade. They were then solvent-washed prior to cellulose extraction using hot acidified $NaClO_2$ followed by sodium hydroxide washes under a nitrogen atmosphere followed by a final acid wash (Norris, 2015; Corran, 2021). It has been demonstrated that this technique is sufficient to remove non-cellulosic material, whereas less rigorous techniques are not (Hua et al., 1999; Norris, 2015). Samples were combusted by elemental analyser or sealed tube combustion (Baisden et al., 2013), IRMS $\delta^{13}C$ measurements were made on a subsampled aliquot of the resultant $CO_2$ gas and the remaining $CO_2$ was graphitised (Turnbull et al., 2015) prior to $^{14}C$ measurement made on our XCAMS system (Zondervan et al., 2015). Standardisation is via Ox-I (Stuiver and Polach, 1977).

Results are reported as $\Delta^{14}C$ (Stuiver and Polach 1977), corrected for decay according to the estimated middle date of each ring growth period. In the Southern Hemisphere where the growth period spans two calendar years, dendrochronologists assign the ring year as the year that growth started, but the average age of the ring is estimated as January 1st of the following year for the purposes of radiocarbon decay correction. Uncertainties are determined from the AMS counting statistics and the repeatability of replicate measurements of the tree ring samples, and is typically 2‰.

Ring counts must be validated to ensure proper chronologies and accurate interpretation of results; this is particularly important since the expected latitudinal gradients are of similar magnitude to the annual $\Delta^{14}CO_2$ trend. We primarily validated ring counts by $\Delta^{14}C$ bomb-spike matching: the rapid changes in $\Delta^{14}C$ during the bomb-spike are so much larger than the spatial variability that a ring count error of a single year is immediately apparent (Andreu-Hayles et al., 2015). Miscounted rings were easily identified by comparing measured $\Delta^{14}C$ to the Baring Head record (examples found in Supplementary Material). Where a ring miscount was identified by this method, we discarded that core from further analysis. Where we were unable to obtain tree cores from a given site that are long enough to include the bomb-spike, ring counts are validated through duplicate sampling and cross-referencing $\Delta^{14}C$ measurements. Further information regarding ring-count validation can be found in the Supplementary Material. Cores from two sites (Baring Head and Eastbourne) used in this work have been previously published (Turnbull et al., 2017)

## 2.2 Development of background reference

To put the tree-ring measurements into context, they are analysed relative to Southern Hemisphere atmospheric background $\Delta^{14}CO_2$ from long term measurements at Baring Head, New Zealand (GNS/NIWA) (Turnbull et al., 2017), and Cape Grim, Tasmania (University of Heidelberg)(Levin et al., 2010), each described below. While modelling studies show lower $\Delta^{14}C$ values over the Southern Ocean attributed to the upwelling of old, $^{14}C$ depleted waters(Graven et al., 2012; Levin et al., 2010), the Baring Head and Cape Grim records from 41°S are not significantly influenced by the Southern Ocean signal (Levin et al., 2010), making them an appropriate anchor point. They are also the most complete and detailed in the Southern Hemisphere, making them a reliable reference.

The Baring Head/Wellington record is the longest-running $\Delta^{14}CO_2$ dataset in the world, and the only direct Southern Hemisphere atmospheric record to capture the $^{14}C$ "bomb-spike". Beginning in 1954, this record has seen changes in methods and sampling sites over the years, discussed in detail in (Turnbull et al., 2017). There is additional noise in the record starting from 1995 when RRL switched methods of $\Delta^{14}C$ measurement from gas-counting to AMS. In 2005, online $^{13}C$ measurement allowed for appropriate fractionation correction, and reduction in this noise (Turnbull et al., 2017; Zondervan et al., 2015). Data in the period between 2009 and 2012 also is significantly offset from the time-series, which is thought to be due to changes in NaOH absorption sampling techniques during this period. These two periods (1995-2005; 2009-2012) are therefore removed from the Wellington Record (Fig 1 (b)).

The University of Heidelberg Institute of Environmental Physics operates a network of time-series stations measuring $\Delta^{14}CO_2$ sampled using the NaOH absorption method (Levin et al., 2010) and analysed via decay counting (Kromer and Munnich, 1992). The Cape Grim, Tasmania station (CGO; 40.68S, 144.68E, (Levin et al., 2010, 2022)) is at a similar latitude to Wellington, and has data available between 1987 and 2018, overlapping the Baring Head record from 1987-2016 (Fig 1 (a/b)).

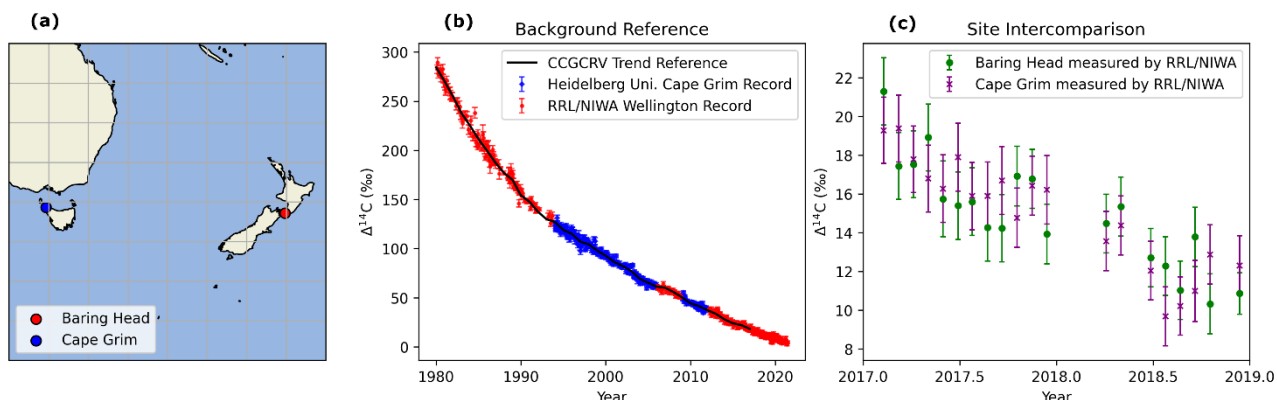

**Figure 1: (a) Locations of Southern Hemisphere atmospheric background records; Cape Grim, Tasmania and Baring Head, Wellington, New Zealand. (b) Atmospheric $\Delta^{14}CO_2$ measured at Baring Head and Cape Grim post-1980.**

**Certain periods of data have been removed, see Methods. (c) Comparison of $\Delta^{14}CO_2$ measured at the two sites by the same laboratory to identify potential spatial driven offsets in atmospheric background. None were found.**

The two time-series' from Baring Head and Cape Grim are combined to fill gaps that exist in both records. Two issues that arise are 1) site-site offsets and 2) $\Delta^{14}C$ measurement offsets. Justification for the merging of two datasets from different sites are 1) Cape Grim and Wellington observe air masses of similar origin (Ziehn et al., 2014), and 2) a two year site-site intercomparison in which air collected from Baring Head and Cape Grim were both processed and measured at NIWA/GNS found no measurable difference between the sites (Fig 1 (c)). Comparability between $\Delta^{14}C$ measurements at RRL and Heidelberg University is within goals re-established by the WMO and GGMT in 2020 (0.5‰) (Crotwell et al., 2020). The merged dataset will subsequently be referred to as the "Southern Hemisphere Background" or SHB.

## 2.3 Interfacing Samples with Reference Dataset

The merged SHB contains discrete temporal values (sampling dates/times), representing the date a whole-air flask sample was collected, or the median sampling date for passive NaOH absorption. The tree-ring $\Delta^{14}CO_2$ data has time values set to January $1^{st}$ (summer) of the growth year. In order to estimate the difference between tree-ring $\Delta^{14}CO_2$ and SHB, the time axis of both datasets must be matched.

We use the NOAA Global Monitoring Laboratory's CCGCRV curve fitting method (Thoning et al., 1989) to interpolate and smooth the SHB for comparison with the tree-ring $\Delta^{14}CO_2$, using a FFT filter cut-off of 667 days. The goal is to reduce the noise associated with the relatively large uncertainties on individual $\Delta^{14}C$ measurements, which can result in a single outlier $\Delta^{14}C$ measurement in the SHB record aliasing into a deviation in $\Delta\Delta^{14}C$ in all the other records. Uncertainties on the smoothed, interpolated data are obtained using a Monte Carlo scheme run to 10,000 iterations as in (Turnbull et al., 2017; Hua et al., 2021). More information about this scheme can be found in the Supplemental Material. The mean of the SHB smoothed data for October to March is used for comparison with the tree-ring $\Delta14CO_2$. Taking the difference between tree-ring $\Delta^{14}CO_2$ and the "trended" SHB yields $\Delta\Delta^{14}CO_2$ (Eq. 1).

$$\Delta\Delta^{14}CO_2 = \Delta^{14}CO_{2\_TreeRing} - \Delta^{14}CO_{2\_SHB} \tag{1}$$

## 2.4 Sampled Time Period

While the tree-ring records in some cases extend back to <1950, and we measured some tree rings from the initial bomb-spike to validate the ring counts, this study focuses on the "Post-bomb period" from 1980 to the 2017 (Turnbull et al., 2016). After the thermonuclear weapons testing created the radiocarbon "bomb-spike", isofluxes (differences between reservoir and atmosphere) from the terrestrial biosphere and the ocean to atmosphere were strongly negative, as the systems were in severe disequilibrium (Levin et al., 2010; Naegler and Levin, 2009; Randerson et al., 2002; Turnbull et al., 2016). We select our samples post-1980 when equilibrium was closer to being established, the sign in biosphere isoflux had changed to positive, and the fossil fuel and ocean signals become the dominant drivers of $\Delta^{14}C$ spatial variability (Levin et al., 2010; Naegler and Levin, 2009; Randerson et al., 2002; Turnbull et al., 2016). Biosphere respiration of bomb $^{14}C$ now had higher $\Delta^{14}C$ than the atmosphere, while the oceans still have a $\Delta^{14}C$ lower than the atmosphere, with the possible exception of some tropical regions (Graven et al., 2012). Fossil fuels always have strongly negative isofluxes, essentially diluting atmospheric $^{14}C$ (Suess 1955; Turnbull et al., 2016; however, in this Southern Hemisphere study we aim to avoid local fossil-fuel contribution.

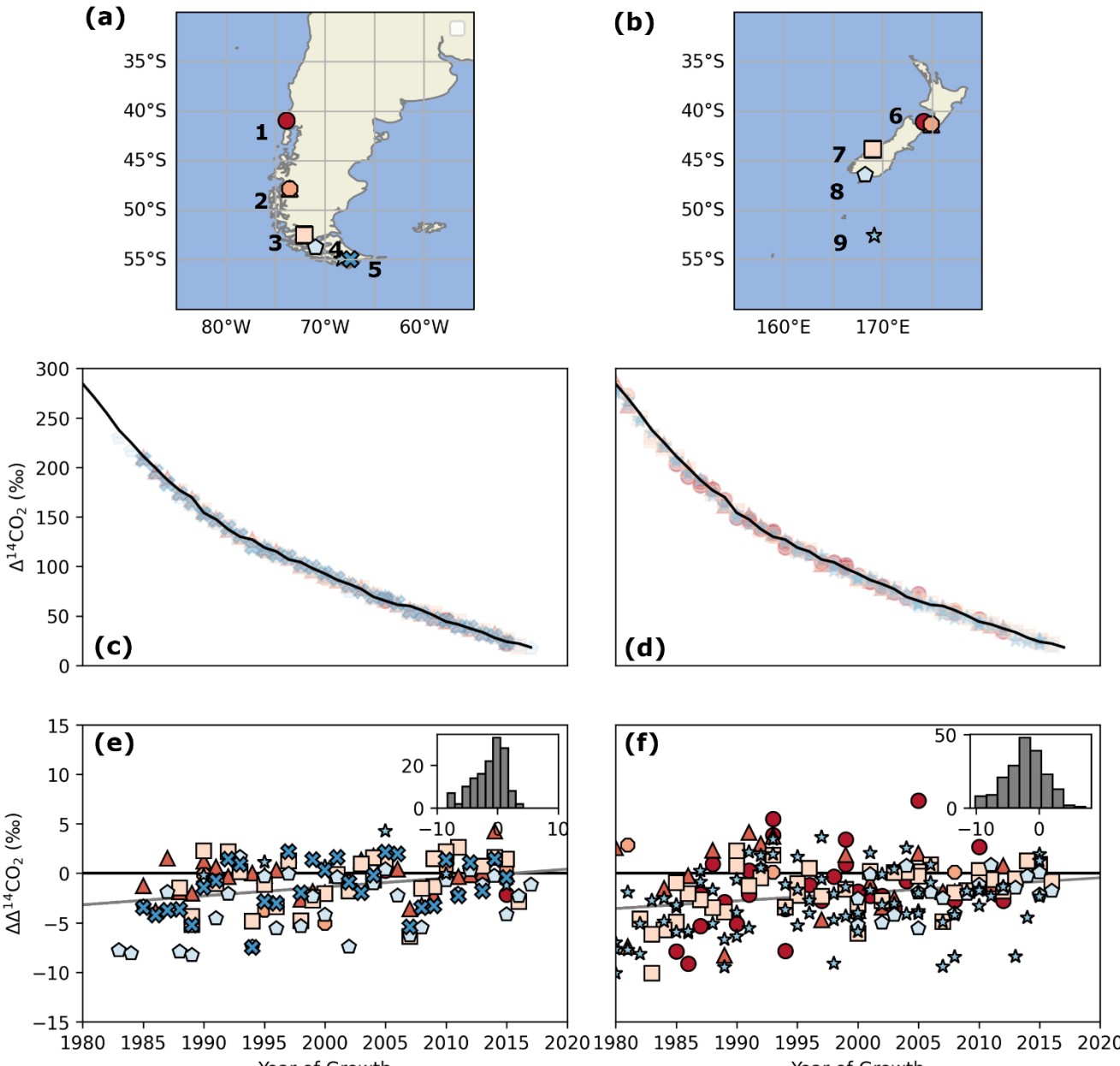

**Figure 2: (a) and (b) show the sampling sites in Chile and New Zealand. Numbers refer to stations as follows: 1) Bahia San Pedro, 2) Tortel River & Tortel Island 3) Seno Skyring 4) Monte Tarn, Punta Arenas, 5) Puerto Navarino & Baja Rosales, Isla Navarino 6) Baring Head & Eastbourne, 7) Haast Beach, 8) Oreti Beach, 9) Campbell Island. (c) and (d) show the measured $\Delta^{14}CO_2$ values overlaid on the Southern hemisphere background smoothed with CCGCRV algorithm. (e) and (f) show the difference between tree ring $\Delta^{14}CO_2$ and the Southern Hemisphere Background ($\Delta\Delta^{14}CO_2$). Inset histograms reflect all data from respective continents binned into 10 equal bins.**

## 2.5 Atmospheric Back-Trajectories

We use the NOAA HYSPLIT air parcel trajectory model (https://www.arl.noaa.gov/hysplit/) to estimate the source regions of air parcels arriving at each sampling site. We determined mean back-trajectories by running HYSPLIT in backwards mode for with starting trajectories once every two days for the three months through the height of the growing season, running backwards in time for six days (December, January, February) for selected years during which meteorological data was available. Years 2005-2006, 2010-2011, 2015-2016, and 2020-2021 were selected. For each site, 185 trajectories were produced. This number of trajectories was selected to give a reasonable representation of the air mass origins over the full time period, while balancing the computational effort required. Results are presented as a heat-map showing the distribution of points over the trajectories. HYSPLIT was run via the Python PySPLIT package (Warner, 2018).

Air mass back-trajectory results are analysed in context to previously constrained Southern Ocean fronts and zones.
We use the subtropical front (STF), subantarctic front (SAF), polar front (PF) and southern boundary (SB) from (Orsi et al., 1995), as well as the climatological PF from (Freeman and Lovenduski, 2016). Climatological fronts from these works were chosen because of their data availability and overlap with the timespan of our tree-ring measurements (1980-2017). Orsi et al., (1995) constrains the STF, SAF, PF, and SB using all available data up to 1990, while Freeman and Lovenduski (2016) PF climatology uses data from 2002 to 2014. Southern Ocean zones are defined as the region between the fronts, similar to
Gray et al., (2018): the subtropical zone (STZ), subantarctic zone (SAZ), polar frontal zone (PFZ), Antarctic-Southern Zone (ASZ), and Seasonal Ice Zone (SIZ) (see Fig. 4). Because our analysis overlaps with the timespans of Orsi et al., (1995) and Freeman and Lovenduski (2016) climatologies, we analysed HYSPLIT back-trajectory results and GLODAP data (see Section 2.5) using both PF positions and report the mean. Changes to the PF position, in our analysis, affects only the PFZ and ASZ results as the remaining fronts are held constant from Orsi et al., (1995).


## 2.5 GLODAP Ocean $^{14}$C data

The GLODAP Merged and Adjusted Data Product v2.2023 is used to add context to the discussion of our results (Key et al., 2004; Lauvset et al., 2023; Olsen et al., 2016). The data was filtered for existing ocean dissolved inorganic carbon radiocarbon measurements ('G2c14') south of 5°S, shallower than 100m, and post-1980 to match our tree-ring temporal span.
Frontal zones from (Orsi, 1995) were interpolated to distinct longitude values of samples in the dataset, to allow comparison of the sample latitude to front latitudes. Then, each measurement was "binned" in one of the Southern Ocean frontal zones, and ocean sector, based on sample latitude/longitudes. Further details on interpolation are included in the Supplementary Materials. The Pacific sector is defined as between 120°E to 70°W, and the Indian sector is between 21°E and 120°E. The mean and standard deviations for nitrate and $\Delta^{14}$C, by ocean sector (and total ocean), for each polar frontal zone are shown in
Figu 6 (c) and (d), and tabulated in the Supplementary Materials.

## 3 Results and Discussion

$\Delta^{14}CO_2$ measurements from each site are overlaid with SHB (Fig 2 c/d) to show an overview of the data. All records reflect the general long-term pattern of decreasing $\Delta^{14}CO_2$ that is observed in the SHB record and globally. Small
deviations from the SHB record become apparent when $\Delta\Delta^{14}CO_2$ is calculated for each site (Fig 2 (e/f)). The inset histogram shows the distribution of all data from the New Zealand or Chilean sector. The most prominent feature of Fig 2 (e/f) is the year-to-year variability at all locations, likely driven by a combination of interannual variability in oceanic $^{14}$C fluxes, atmospheric transport, and tree growth periods. This manuscript intends to evaluate latitudinal gradients only. Temporal trends will be discussed further in a companion manuscript. Raw $\Delta^{14}CO_2$ data (Fig 2(e/f)) are not temporally de-trended
before means are taken (data in Fig 3).

### 3.1 Comparison with other Southern Hemisphere Atmospheric $\Delta^{14}$C Records

The measurements presented here expand the Southern Hemisphere records of atmospheric $\Delta^{14}$C previously described in Levin et al., (2010) and Hua et al. (2021). We briefly compare these new data to existing Southern Hemisphere
$\Delta^{14}$C datasets at Cape Grim, Australia (Levin et al., 2010), Campbell Island (Manning et al., 1990; Turney et al., 2018), Macquarie Island (Levin et al., 2010) and Neumayer Station (Levin et al., 2010).

In Turnbull et al., (2017), the authors publish Baring Head and Eastbourne tree-ring $\Delta^{14}$C measurements, and provide a rigorous comparison to show that the Wellington $\Delta^{14}CO_2$ record agrees well with atmospheric measurements from Cape Grim, Australia (Levin et al., 2010, 2022). An additional two-year site-site intercomparison provides confidence that
Cape Grim and Baring Head are not different (Fig 1. (c)).

Manning et al., (1990) and Turney et al., (2018) report atmospheric and tree-ring measurements from Campbell Island, respectively. The atmospheric record from Manning et al., (1990) extends from 1970-1977 and does not overlap with our tree-ring measurements ruling out a direct comparison with our Campbell Island tree-rings. Turney et al., (2018) includes tree-ring measurements from the same *Sitka spruce* tree, but the record only extends to 1967, disallowing a direct

comparison. The *Dracophyllum* records extend from 1952 to 2011, overlapping well with our record. If the Turney et al., (2018) *Dracophyllum* records are analyzed according to our workflow, we find there is a robust difference ($p<0.05$) between it and the Campbell Island record presented in this work (see Supplementary Materials), with Turney et al., (2018) offset higher. Fossil fuel $CO_2$ would drive $\Delta^{14}CO_2$ lower, but is highly unlikely due to the absence of fossil influence in the region (Levin et al., 2010) and the uninhabited nature of the island. Biospheric contamination leading to a relative increase of the

Turney et al., (2018) data is similarly unlikely due to sustained high westerly winds. We think the most likely explanation is an interlaboratory offset. However, an important consideration is that the records and results we show were all sampled, processed, and measured identically, meaning results from Campbell Island are best viewed relative to other sites from this work. Additionally, low values at Campbell Island can be validated by matching the Macquarie Island record (see below), where interlaboratory offsets between the two institutions have been investigated and none found (see Baring Head versus

Cape Grim above and Hammer et al., (2017)).

It is instructive to compare Campbell Island (52.5S, 169.2E) to the nearby Macquarie Island atmospheric record (54.6S, 158.9E) from 1992 to 2004 (Levin et al., 2010). Fig. 4 shows direct comparisons of the sites' $\Delta^{14}CO_2$ and $\Delta\Delta^{14}CO_2$, the latter if the Macquarie Island record is analysed in our workflow. The Macquarie Island atmospheric record is -2.6±3.2‰ lower than background, similar to Campbell Island $\Delta\Delta^{14}C$. Independent t-test also shows the datasets are not different ($p =$

0.3). This specific comparison provides confidence that the result we find at Campbell Island is robust. Macquarie Island is previously the only record at this latitudinal band to capture low $\Delta^{14}C$ over the Southern Ocean. The new Campbell Island record supports the original Macquarie Island record but also extends is back to 1980 and forward to 2017.

Levin et al, (2010) found that Macquarie Island and Neumayer Station atmospheric records (70.4S, 8.2E) are both influenced by Southern Ocean outgassing. Comparing Neumayer Station and Campbell Island shows there is no robust

difference between the sites ($p=0.5$).


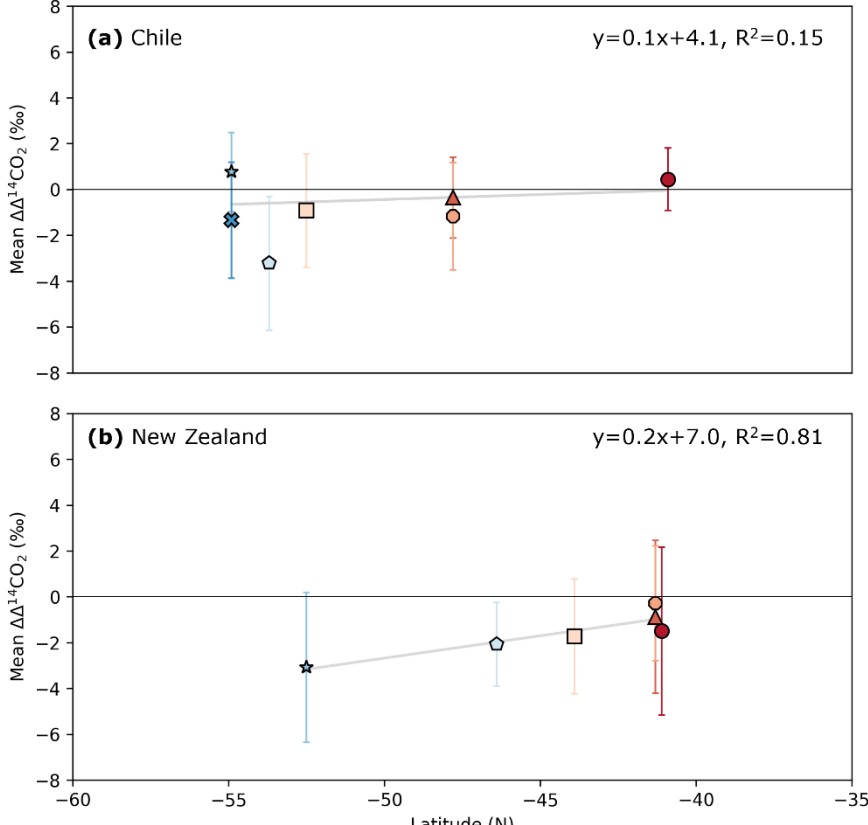

Figure 3: Mean $\Delta\Delta^{14}CO_2$ value for each individual site in (a) Chile and (b) New Zealand. Symbols and colors correspond to Fig 2. Error bars represent the 1 standard deviation of the mean of each site's dataset of $\Delta\Delta^{14}CO_2$ measurements; with each individual measurement error propagated throughout the analysis. A version of this figure with individual $\Delta\Delta^{14}CO_2$ values and 95% confidence intervals overlaid can be found in the Supplementary Materials.

**3.2 New Zealand Tree Ring $\Delta^{14}C$**

The mean and one standard deviation of $\Delta\Delta^{14}CO_2$ for each site over the record is shown in Fig 3. The colors and symbols correspond to Fig 2. The three sites at or near Baring Head (origin of background record) (Fig 3(b)) are close, but not identical, to SHB. Tree-rings incorporate air over a months-long growing season regardless of wind-direction, while individual measurements used for SHB are either a temporal snapshot (air flask sample) or a two-week integration (NaOH absorption). This difference is a likely driver of such variability.

All New Zealand sites' $\Delta\Delta^{14}CO_2$ means are within 1-σ of 0‰ besides Oreti Beach (46°S). Nonetheless, there is a linear trend ($R^2$ = 0.81; p = <<0.05) and steeper slope compared to that of Chilean sites. This slope is driven by our southernmost New Zealand site, Campbell Island, with lowest mean $\Delta\Delta^{14}CO_2$ (−3.1 ± 3.3‰). Because our results are not temporally detrended before the mean is taken, if Campbell Island tree-rings indeed capture decadal variability in atmospheric $\Delta^{14}C$ forced by changes in Southern Ocean outgassing, this will increase the variability in the record. This will be further explored in future work.

Results from HYSPLIT back-trajectories are displayed as heatmaps for selected sites in Fig 5, with remaining sites shown in the Supplementary Material. Heatmap data is also displayed as a bar chart in Fig 6, with a table in the Supplementary Material. Campbell Island is the only New Zealand site where the HYSPLIT back-trajectory plume core sits in the SAZ. HYSPLIT modelling indicates that air moving toward Campbell Island spent the most amount of time in both the PFZ and ASZ (15.3 and 24.4%; Figure 5). Most Southern Ocean outgassing occurs in these two zones, with the latter showing near year-round outgassing (Gray et al., 2018).

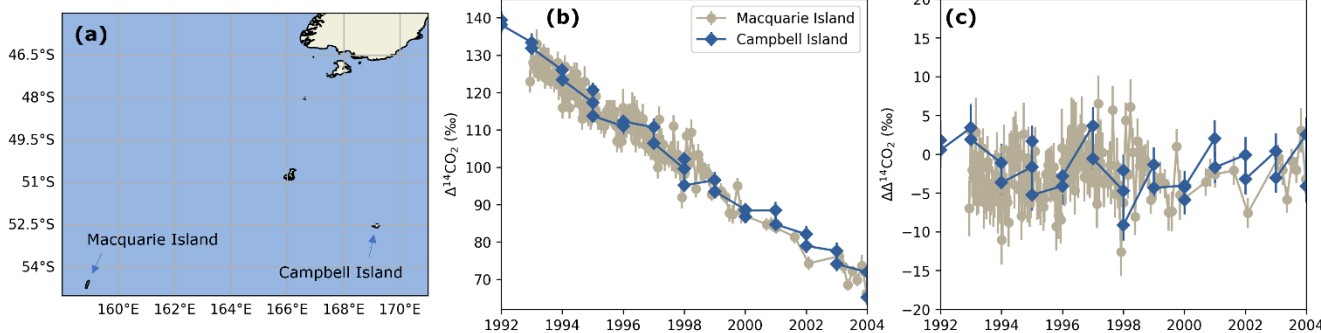


**Figure 4: (a) Proximity of Macquarie Island (Levin et al., 2010) and Campbell Island. (b) $\Delta^{14}CO_2$ and (c) $\Delta\Delta^{14}CO_2$ from Campbell Island and Macquarie Island.**

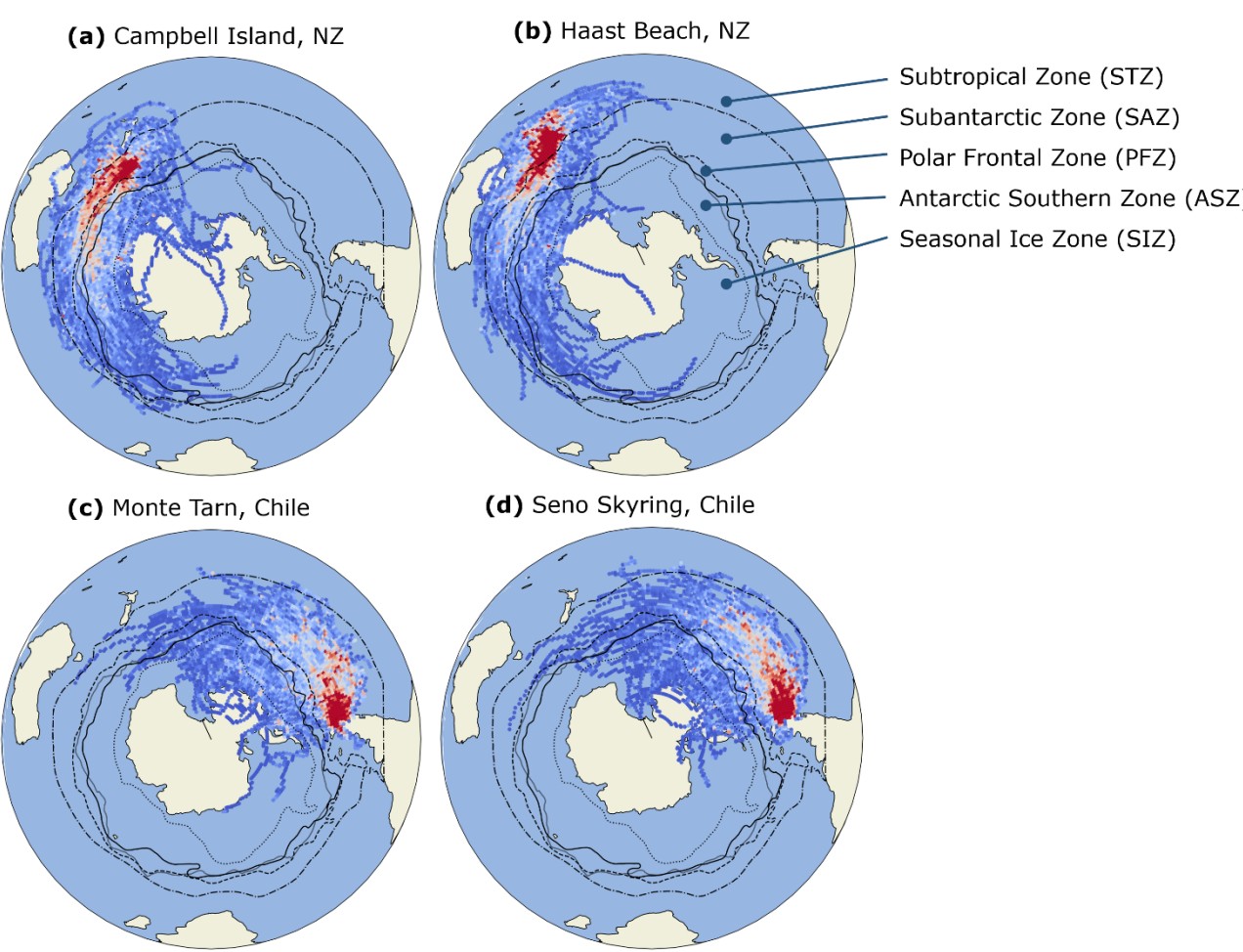


**Figure 5: HYSPLIT back-trajectories shown for select sites. Trajectories for all other sites are found in Supplementary Material. Southern Ocean fronts presented in bold are from (Orsi et al., 1995) and semi-transparent polar front from (Freeman and Lovenduski et al., 2016) climatology.**

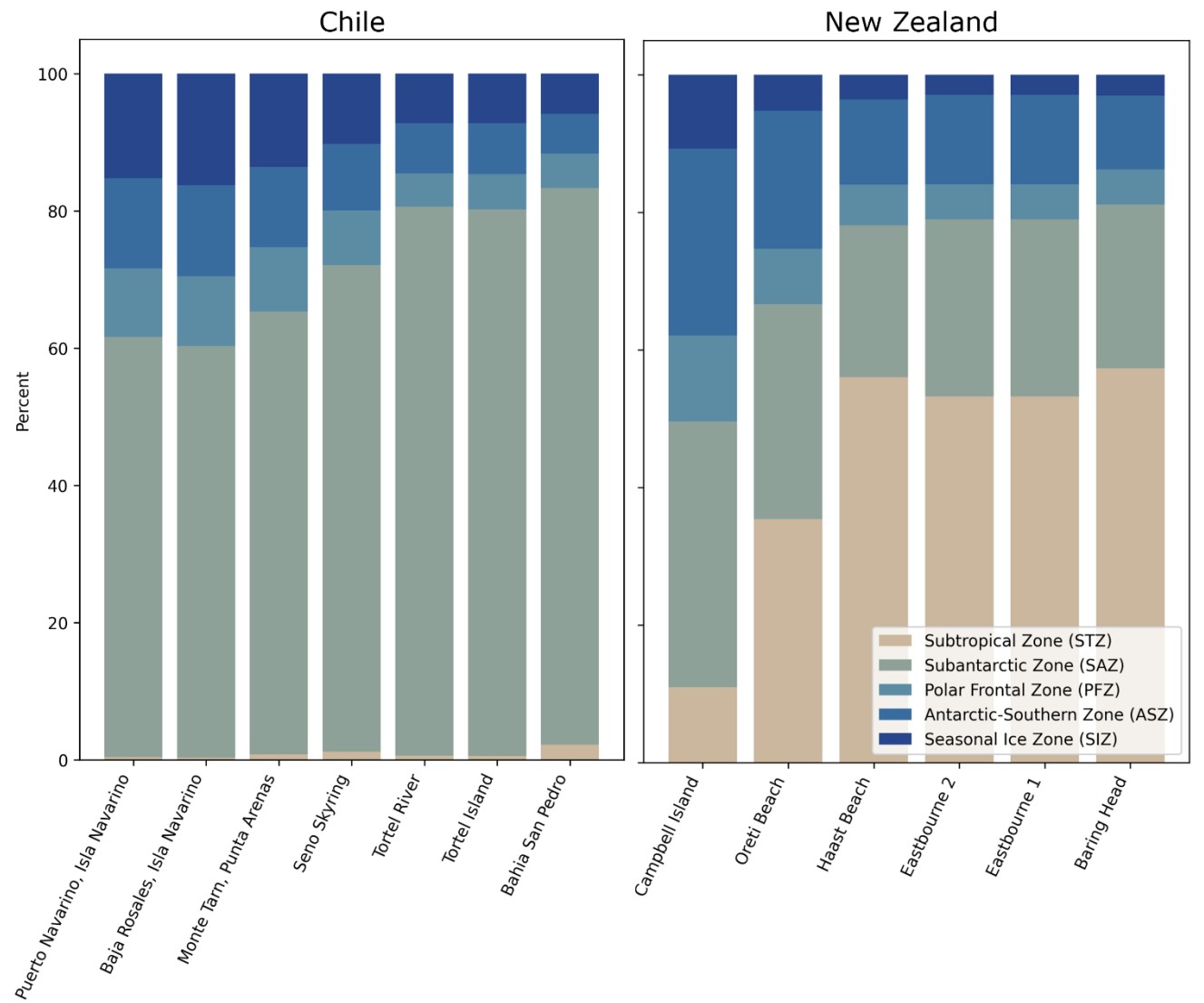

**Figure 6: Visual representation of the amount of time HYSPLIT back-trajectories from each site spend in each Southern Ocean frontal zone.. PFZ and ASZ represent the mean of values calculated using PF climatology from (Orsi et al, 1995) and (Freeman and Lovenduski, 2016).**


Figure 7 shows nitrate and DIC $\Delta^{14}$C from GLODAP (see Methods). Nitrate increases and DIC $\Delta^{14}$C values decrease toward higher latitudes into the PFZ, ASZ, and SIZ. DIC $\Delta^{14}$C decreases by ~150‰ between the STZ and ASZ. These are distinct markers not only of Southern Ocean upwelling but of the low $\Delta^{14}$C values imparted from old surface water masses to overlying air-masses carried on to our tree-ring sites. We hypothesize that the New Zealand $\Delta\Delta^{14}CO_2$ latitudinal

gradient is driven by Southern Ocean outgassing in the PFZ and ASZ which feed air to Campbell Island in greater abundance relative to more northern sites. Campbell Island tree-rings may therefore be useful for detecting changes in outgassing in the Indian Ocean sector of the Southern Ocean PFZ and ASZ.

**3.3 Chilean Tree Ring $\Delta^{14}C$**

A weak latitudinal gradient exists in the Chile dataset; however, this is driven by one site (Monte Tarn; 53.7°S). Excluding this site, all means are indistinguishable from SHB, and the linear regression slope falls from 0.1 to 0.03, and $R^2$ from 0.15 to 0.05 (p=0.4). Including all data, the Chilean trend has p-value 0.395, and is not statistically significant.

Monte Tarn has the lowest mean $\Delta\Delta^{14}CO_2$, is the only site outside 1-σ of the SHB (mean= −3.2 ± 2.9‰) and, notably, is not the most southerly Chilean site. It is also shrouded by the mountainous barrier island Isla Clarence, and is nearby a shipping lane leading to Punta Arenas. Sites in this study were specifically chosen on western coastlines to maximize the amount of clean ocean-air originating from predominant westerlies around the ACC. This makes the likelihood of fossil fuel $CO_2$ incorporation into the tree-rings low. Fossil $CO_2$ incorporation would lower $\Delta\Delta^{14}C$ in Monte Tarn tree rings; however, it is difficult to compose an estimate of the magnitude due to the privacy of shipping data.

Our southernmost Chilean records are two sites on Isla Navarino at the same latitude of 54.9°S. Of the two, Puerto Navarino lies further west and is in proximity to the Argentinian city of Ushuaia, while Baja Rosales is 0.9° to the east. These two sites were selected with the expectation that any significant land biosphere signal or fossil fuel emissions from urban influence would lead to measurable differences between the two sites. However, no statistically significant offset is found between them (see Supplemental Material). Mean $\Delta\Delta^{14}CO_2$ of both sites are within 1-σ of zero (Puerto Navarino: 0.8± 1.7‰; Baja Rosales: -1.3±2.5‰).

**3.4 Ocean Sector Influence on Atmospheric $\Delta^{14}CO_2$**

We hypothesize that steeper latitudinal gradients are found in New Zealand versus Chilean sites due to variability in the spatial extent of Southern Ocean zones in the Indian and Pacific Oceans, and differences in $\Delta^{14}C$ values of DIC upwelling in the ASZ in the Indian versus Pacific sectors.

Southern Ocean carbon flux is complex and variable, with different sectors acting as sources and sinks for anthropogenic and natural $CO_2$, respectively (Gruber et al., 2019). The Southern Ocean as a whole acts as a sink for anthropogenic CO2, with uptake concentrated in the PFZ (Mikaloff Fletcher et al. 2006, DeVries 2014; Gruber et al., 2019). In the case of natural CO2, higher latitudes, specifically those of the ASZ and PFZ, dominate outgassing while lower latitudes such as in the STZ, dominate uptake (Gruber et al., 2019). Some zones outgas seasonally, while the ASZ outgasses nearly year-round (Gray et al., 2018). Uptake and outgassing are also zonally asymmetric. The Indian and Pacific sectors jointly dominate Southern Ocean outgassing. These complexities play key roles in facilitating a signal of one-way flux of low $\Delta^{14}C$ from the surface ocean toward distant trees.

Hypothetically, if the amount of $CO_2$ originating from the same proportion of high-latitude Southern Ocean zones (SIZ, ASZ, PFZ) to Chile and New Zealand were equal, it is reasonable to assume that our method should find the same gradient in $\Delta\Delta^{14}CO_2$ with latitude for Chile and New Zealand tree-rings. The Pacific has higher DIC and lower $O_2$ concentrations in Indian Pacific Deep Waters, which upwell around the ACC band (Chen et al., 2022), suggesting it is more carbon rich and old. The Pacific also dominates outgassing in the PFZ (Prend et al., 2022) With these conditions, one would expect lower $\Delta\Delta^{14}CO_2$ values in Chilean tree-rings than New Zealand tree-rings.

Despite this, we find a stronger outgassing influence in New Zealand, which lies east of the Indian Ocean. This discrepancy is especially prominent if the Chilean site Monte Tarn is excluded due to its proximity to a shipping lane, leaving Campbell Island with the lowest $\Delta\Delta^{14}CO_2$ value.

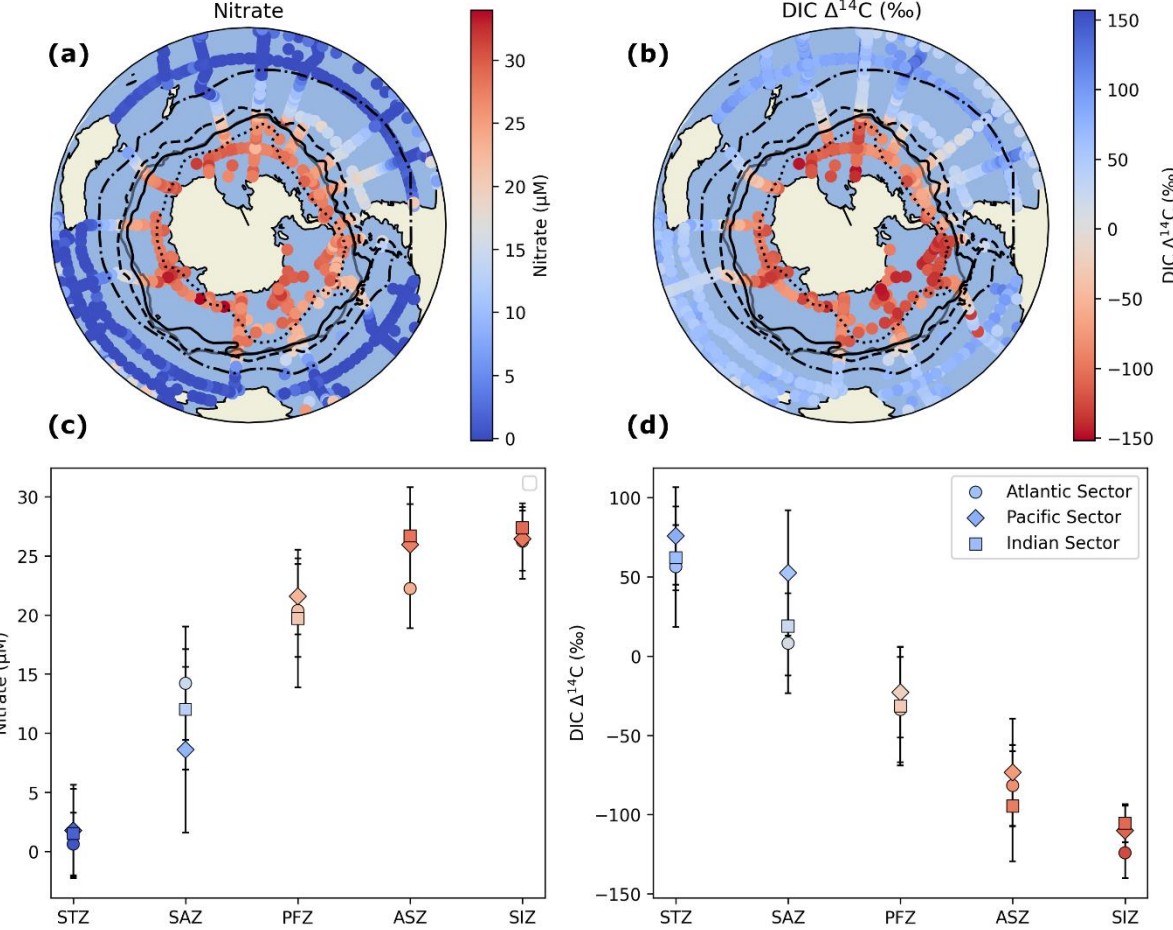

**Figure 7: (a/b) Nitrate and DIC Δ¹⁴C data from GLODAP Merged and Adjusted Data Product v2.2023** (Key et al., 2004; Lauvset et al., 2023; Olsen et al., 2016)**. Data were filtered for measurements south of 5S, and depths 0-100m, after 1980. (c/d) The mean and 1-σ standard deviation of nitrate DIC Δ¹⁴C for each ACC zone (Southern Ocean fronts presented in bold are from (Orsi et al., 1995) and semi-transparent polar front from (Freeman and Lovenduski et al., 2016) climatology.In (c) and (d), PFZ and ASZ represent the mean of values calculated using PF climatology from (Orsi et al, 1995) and (Freeman and Lovenduski, 2016).**

We hypothesize the driving factor is that a larger proportion of New Zealand back-trajectories lie in the crucial year-round outgassing ASZ because the zone is expanded in the Indian Ocean sector relative to Pacific. The ASZ surface also has lower Δ¹⁴C than more northerly zones (Fig 6d) amplifying the effect of atmospheric dilution of ¹⁴C relative to outgassing in general. Due to Campbell Island trees ability to capture dilution in atmospheric Δ¹⁴C from Southern Ocean outgassing, tree rings from this site may be a viable candidate to reconstruct changes in Southern Ocean outgassing in the past few decades.

**4 Conclusion**

While other methods of estimating Southern Ocean air-sea $CO_2$ flux face the challenge of distinguishing between two large opposing forces (outgassing of natural $CO_2$ and uptake of natural $CO_2$), radiocarbon measurements through long-term atmospheric records (Levin et al., 2010) or tree-rings allow us to constrain a one-way flux by measuring dilution of atmospheric radiocarbon from outgassing of $CO_2$ from aged water masses. They also provide the opportunity to reconstruct changes in the past when hydrographic and float-based data were sparse. We report a novel database of 280 unique tree ring Δ¹⁴C measurements from Chile and New Zealand from 1980-2017. This work substantially expands the Δ¹⁴C records from the Southern mid-high latitudes and is consistent with previous studies that demonstrate the imprint of Southern Ocean

upwelling on atmospheric $\Delta^{14}C$ in the Southern Hemisphere(Levin et al., 2010; Graven et al., 2012). The upwelling signal is most apparent at Campbell Island, the southernmost New Zealand site that has HYSPLIT back-trajectory footprints in the ASZ, the only ocean zone to exhibit year-round outgassing (Gray et al., 2018). The link between low $\Delta\Delta^{14}CO_2$ in Campbell Island tree-rings and air mass origination in the Indian Sector ASZ should be further explored for viability as a proxy for detecting changes in upwelling.

Data at more northerly sites and those that are influenced by air masses from lower-latitude ocean zones appear fairly homogenous. This suggests that the influence of non-local fossil fuel and biospheric signals are small in the Southern Hemisphere. Further investigation is required to understand the mechanism behind low values at Chilean site Monte Tarn, and if fossil fuel contamination from a nearby shipping lane may play a role.

Tree-ring $\Delta^{14}C$ measurements are key tools to reconstruct atmospheric $\Delta^{14}C$ and may provide new opportunities when trying to understand changes in Southern Ocean upwelling, and air-sea $CO_2$ flux. This work also highlights the potential for ship-based atmospheric $\Delta^{14}C$ measurements to detect changes in ocean upwelling. Ship-based atmospheric $\Delta^{14}C$ samples can be collected with higher temporal frequency, without the need for oceanographic research vessels. More investigation is required to understand *temporal* changes in $\Delta\Delta^{14}CO_2$. In a companion work to follow, we will address trends over time and analyse how our trends compare with (Le Quéré et al., 2007) and ocean model output.

**5 Code availability**

Scripts used to create this work can be found in the GitHub directory below. Supplemental Materials includes a detailed list of the job each script performs in the data-analysis workflow. The GitHub page also includes a list of dependencies required for the scripts to function.

https://github.com/christianlewis091/science_projects/tree/main/soar_tree_rings/scripts_EGU_REVIEW

**6 Data availability**

Tree ring $\Delta^{14}C$ measurements (Fig. 2 (e/f)), mean $\Delta\Delta^{14}CO_2$ (Fig. 3), results from HYSPLIT back-trajectory modelling (Fig. 5), and summary results from GLODAP analyses (Fig. 6) are available at **10.5281/zenodo.15192463** in .xlsx format. Comments above data on each tab reference codes in GitHub directory described above. Data used for GLODAP analysis is publicly available at https://glodap.info/index.php/merged-and-adjusted-data-product-v2-2023/. Baring Head atmospheric time series data can be found as described in Data Availability statement in (Turnbull et al., 2017), and Cape Grim time series data described in (Levin et al., 2010).

**7 Supplement**

Supplemental material is available online at {link to be determined after submission}.

**8 Author contributions**

**RC:** conceptualization, sample collection, sample processing, analysis and writing; **SMF** and **EB**: analysis and interpretation, methodology, and writing; **RM** and **GB**: maintenance and sample curation from Baring Head Atmospheric Research Station and site-site intercomparison; **AL**: administration, methods, and supervision of tree-ring processing, **MN**: administration. methodology, resources, writing; **CBL**: formal analysis, investigation, methodology, software, validation, visualization, writing; **JT**: funding acquisition, conceptualization and data curation, analysis, methodology, supervision and writing. All authors were involved in reviewing and editing the manuscript.

## 9 Competing Interests

The authors declare that they have no conflict of interest.

## 10 Acknowledgements

The authors would like to acknowledge Cameron Johns, Bjorn Johns, Malcolm Turnbull, Ian Turnbull and Jane Forsyth for advice and participation regarding tree-ring sampling in New Zealand's South Island. We also acknowledge Dave Bowen and Alex Fergus of Heritage Expeditions, and cruise members Edin Whitehead, Paul Charman, and Hamish Sutherland for their assistance with tree-core collection in New Zealand's subantarctic Islands. For assistance with fieldwork in Chile, we thank Carolyn McCarthy, Vince Beasley, Ricardo de Pol-Holz, Juan Carlos Aravena, and Guillermo Duarte. We thank the technicians and scientists of the Rafter Radiocarbon Laboratory and XCAMS for their assistance in sample preparation and radiocarbon measurement.

## 11 Financial Support

This work is supported by the Ministry of Business Innovation and Employment of New Zealand and the Antarctic Science Platform.

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
