# Peer review of "Southern Hemisphere tree-rings as proxies to reconstruct Southern Ocean upwelling"

_EGUsphere, 2024_

## Author Comment (AC2)

Dear Reviewer,

On behalf of the co-authors and myself we appreciate your thoughtful feedback. We submit the following responses for consideration. Your comments are italicized in gray and my responses are in black.

Best wishes,

Dr. Christian B. Lewis

*This study lays the groundwork for the use of Southern Hemisphere tree-rings for studying changes in Southern Ocean upwelling and air-sea $CO_2$ flux. The authors demonstrate that trees from Chile and New Zealand from the 1980s to present accurately record atmospheric 14C (with a few exceptions) with a latitudinal gradient observed for New Zealand sites. Using HYSPLIT software they model back-trajectories for the air masses reaching the site and show that the New Zealand gradient is driven by Southern Ocean outgassing and atmospheric transport. The methods used are robust and the manuscript is generally well-written. Due to the importance of the Southern Ocean in global climate, this study will be critical for future work on constraining temporal changes in upwelling and air-sea gas exchange.*

*Minor comments: (strikethrough indicates completion/comment is resolved)*

*Line 30: Insert space after (Talley, 2013)* Done

*Line 33: Possibly insert 'content' after 'carbon' just to be clear that this doesn't refer to 14C (although probably fine).* Done

*Table 1: 'The final column references numbered locations on Figure 2a/b maps.' Note: The final column with location number is missing.* Done

*Line 163: 'using a FFT filter cut-off of 667' What are the units?*
The value for fc is specified in 'number of days'. Figure 4 shows the response function with cutoff values of 80 days for the short term filter and 667 days for the long term filter. https://gml.noaa.gov/ccgg/mbl/crvfit/crvfit.html

*Line 168: 'Data will be included in Supplementary Material' Is this yet to be done or will it be at 10.5281/zenodo.14532802?*

I've simply removed this line because it may cause confusion. I will let readers find the Data Availability statement rather than bring it up piecemeal throughout the paper.

*Figure 2: Perhaps change the label 'CCGCRV Trend Reference' to 'CCGCRV SHB Trend' for clarity*

It may be clearer to simply remove the legend. An extra line was added in Figure caption to say the line has been smoothed.

*Line 209: 'The data was filtered for existing radiocarbon measurements ('G2c14')' Please insert 'ocean' in front of radiocarbon.*

Following text was edited:
The GLODAP Merged and Adjusted Data Product v2.2023 is used to add context to the discussion of our results (Key et al., 2004; Lauvset et al., 2023). The data was filtered for existing ocean dissolved inorganic carbon radiocarbon measurements ('G2c14') south of 5°S, shallower than 100m, and post-1980 to match our tree-ring temporal span.

*Line 213: 'on bottle latitude/longitudes'  Some readers may not be familiar with the use of bottles for ocean water collection- perhaps replace with 'sample collection latitude/longitudes'*

Fixed

*Line 309: '(Prend et al., 2022) discovered a deep maximum mixed layer depth surrounding Campbell Island and westward in the ASZ (Prend 2022 Fig. 1c).'  The sentence should start with 'Prend et al. (2022) discovered' and presumably end with (Prend et al. 2022, Fig. 1c)*

Fixed

*Line 313: 'Low ΔΔ14CO2 at Campbell Island may be linked air originating'. Insert 'to' after 'linked.*

Fixed

*Line 326: 'local effects such as nearby obduction from deep mixed layer' Doesn't obduction refer to crustal movements?*
Prend et al. (2022) define obduction as follows:
"We find that the interbasin variation in surface ocean pCO2 is primarily driven by regional variability in wintertime entrainment of carbon-rich deep water from the permanent pycnocline ("obduction")." (Prend et al., 2022).

We have removed this term to avoid further confusion.

*Line 329: 'This suggests that the influence of non-local fossil fuel and biospheric signals are small in the Southern Hemisphere.'  Does this apply to the entire Southern Hemisphere?*

Yes. This response in in alignment with previous observational and modelling studies that show the latitudinal gradients of 14C from pole-to-pole suggests that fossil fuel emissions and biosphere signals are small in the Southern Hemisphere. 14C increases in the tropics due to lower fossil fuel emissions and biospheric signals, but decrease in the Southern Hemisphere due to Southern Ocean upwelling before increasing again

toward the pole due to stratospheric production of 14C ((Graven et al., 2012; Levin et al., 2010; Randerson et al., 2002; Turnbull et al., 2009; Turnbull et al., 2017)).

*The citations need to be carefully checked as many in the text are missing from the reference section or incomplete.  I've found the following problems with citations but may have missed some:* Fixed

Citations not in reference section:

References

Graven, Heather D., Nicolas Gruber, Robert Key, Samar Khatiwala, and Xavier Giraud. "Changing controls on oceanic radiocarbon: New insights on shallow-to-deep ocean exchange and anthropogenic CO2 uptake." *Journal of Geophysical Research: Oceans* 117, no. C10 (2012).

Key, Robert M., Alex Kozyr, Chris L. Sabine, Kitack Lee, Rik Wanninkhof, John L. Bullister, Richard A. Feely, Frank J. Millero, Calvin Mordy, and T-H. Peng. "A global ocean carbon climatology: Results from Global Data Analysis Project (GLODAP)." *Global biogeochemical cycles* 18, no. 4 (2004).

Lauvset, Siv K., Nico Lange, Toste Tanhua, Henry C. Bittig, Are Olsen, Alex Kozyr, Marta Álvarez et al. "The annual update GLODAPv2. 2023: the global interior ocean biogeochemical data product." *Earth System Science Data* 16, no. 4 (2024): 2047-2072.

Levin, Ingeborg, Tobias Naegler, Bernd Kromer, Moritz Diehl, Roger Francey, Angel Gomez-Pelaez, Paul Steele, Dietmar Wagenbach, Rolf Weller, and Douglas Worthy. "Observations and modelling of the global distribution and long-term trend of atmospheric 14CO2." *Tellus B: Chemical and Physical Meteorology* 62, no. 1 (2010): 26-46.

Prend, Channing J., Alison R. Gray, Lynne D. Talley, Sarah T. Gille, F. Alexander Haumann, Kenneth S. Johnson, Stephen C. Riser, Isabella Rosso, Jade Sauvé, and Jorge L. Sarmiento. "Indo-Pacific sector dominates Southern Ocean carbon outgassing." *Global Biogeochemical Cycles* 36, no. 7 (2022): e2021GB007226.

Randerson, J. T., I. G. Enting, E. A. G. Schuur, K. Caldeira, and I. Y. Fung. "Seasonal and latitudinal variability of troposphere Δ14CO2: Post bomb contributions from fossil fuels, oceans, the stratosphere, and the terrestrial biosphere." *Global Biogeochemical Cycles* 16, no. 4 (2002): 59-1.

Turnbull, Jocelyn C., Sara E. Mikaloff Fletcher, Gordon W. Brailsford, Rowena C. Moss, Margaret W. Norris, and Kay Steinkamp. "Sixty years of radiocarbon dioxide measurements at Wellington, New Zealand: 1954–2014." *Atmospheric Chemistry and Physics* 17, no. 23 (2017): 14771-14784.

Turnbull, Jocelyn, Peter Rayner, John Miller, Tobias Naegler, Philippe Ciais, and Anne Cozic. "On the use of 14CO2 as a tracer for fossil fuel CO2: Quantifying uncertainties using an atmospheric transport model." *Journal of Geophysical Research: Atmospheres* 114, no. D22 (2009).

---

## Author Response (AR1)

**RC1**: 'Review', Anonymous Referee #1, 27 Jan 2025  reply

*This paper presents new 14C data from tree rings in New Zealand from the last 45 years or so. I believe the data are worth publishing, but I have a few concern on the present state of the draft, which might need improvements:*

1. *In my view the new data should be set in a better context of existing 14C data of the southern hemisphere (SH) of the last 100 years or so. While the authors cite Hua et al 2021 (which should be cited as published in 2022, https://doi.org/10.1017/RDC.2021.95), which  split (following earlier papers) the 14C data 1950-2019 into 5 zones (2 in the SH), I am missing how the new data would agree/disagree with SH zone 1-2 (which is where the positions of the trees from which the new 14C data are measured are situated). Furthermore, Levin et al. (2022, https://doi.org/10.1017/RDC.2021.102) also showed 14C data for different latitudes. Here, especially Neumeyer Station in Antarctica and Macquary Island in the Southern Ocean are of interest. So, at best, the new data from Lewis et al are presented in the wider context of the data presented in these two papers. In this context it might be necessary to recalculate the Southern Hemisphere atmospheric background of Δ14CO2, from which the anomalies of individual sites are calculated.*

As (Hua et al., 2021) is a compilation, I can directly compare my work with the individual publications *within* the compilation. I will explore this moving north to south, beginning with the latitude of Wellington (41°S).

**Latitude 41S: Wellington vs Cape Grim Australia**

Something we did not adequately make clear in the first version of this manuscript is that tree-cores from Baring Head and Eastbourne from which $\Delta\Delta^{14}CO_2$ is calculated *are* tree cores from Turnbull et al. (2017), which Hua et al., (2021) incorporates into their manuscript. We will make this clear in the subsequent version. In Turnbull et al. (2017), the authors discuss how the Wellington and Cape Grim records are "generally consistent with one another" despite a period of time where instrument difficulties led to anomalous values at the Rafter Radiocarbon Laboratory (RRL). In Turnbull et al. (2017) and in this work, that period of time is removed and replaced with the University of Heidelberg Cape Grim record for the Southern Hemisphere Background (Levin et al., 2010).

We also present a direct site-site intercomparison in which Cape Grim and Baring Head are directly compared and with $\Delta^{14}CO$ measured in the same laboratory to test interlaboratory offsets. Although the temporal range is short, we also find no difference

between the site (Fig. 1(c).). These lines of evidence provide confidence that there is no difference between the Wellington and Cape Grim records.

**Campbell Island 52.5°S.**

*Atmospheric records* exist on Campbell Island but extend from 1970 to 1977 (Manning et al., 1990), ruling out a direct comparison with our Campbell Island *tree-ring record*.

Another Campbell Island tree-ring record exists (Turney et al., 2018) which includes records from the same *Sitka spruce* and *Dracophyllum*. The *Sitka spruce* record only extends to 1967, disallowing a direct comparison with our records from the same tree.

If the public access *Dracophyllum* records are analyzed according to our workflow, we find there is a difference between the *Dracophyllum* record (Turney et al., 2018) and the Campbell Island record presented in this work (see Fig. R1(c)). Independent t-test of $\Delta\Delta^{14}C$ yield a p-value of <<0.05, indicating that the two datasets are robustly different. Visual inspection shows the (Turney et al., 2018) records seem to deviate from our record around 1990 and 2007. Even if these periods of deviation are removed from the t-test, the data are still robustly different. Mean $\Delta\Delta^{14}C$ (from our workflow) for (Turney et al., 2018) data is 0±2.6‰ (no difference from Southern Hemisphere background) while our record shows Campbell Island is -3.1±3.3‰ from background.

What would lead to different results from tree-ring measurements on the same island? Fossil fuel contamination would push $\Delta^{14}C$ values lower, while biospheric $CO_2$ would increase $\Delta^{14}C$. Since the island is uninhabited, there is little chance of fossil fuel contamination decreasing our results; however, sustained high westerly winds also make biospheric contamination in (Turney et al., 2018) similarly unlikely. One additional source of uncertainty is that *Dracophyllum* and *Sitka spruce* have different tree-ring widths and potentially different growth periods which may lead to differences in records from differing species, even from the same island.

One key aspect of the work we present is that Campbell Island is compared to other locations which were sampled and processed the same way, by the same people, and measured in the same AMS system. This means that Campbell Island record is best viewed *relative* to sites from the same work.

**Macquarie Island and Neumayer Station**

We will add in the next manuscript version a direct comparison between our Campbell Island record and the (Levin et al., 2010) Macquarie Island record (see Fig. R1(d,e)). This comparison is instructive because they are in close proximity. Independent t-tests confirm that the Campbell Island tree-rings and the Macquarie Island atmospheric measurements are not statistically different (p = 0.3). Macquarie Island record (if put through our analytical workflow) yields -2.6±3.2‰, similar to Campbell Island $\Delta\Delta^{14}C$.

This agreement between Macquarie Island and Campbell Island demonstrates that our result is realistic, and justifies our use of a longer Campbell Island record.

Neumayer Station similarly is found to be no different from Campbell Island (p=0.5) with a mean $\Delta\Delta^{14}C$ of -2.7±5.2‰.

[Figure]

**Fig R1.** A comparison of Campbell Island tree-ring $\Delta^{14}C$ with other Southern Hemisphere sites including another Campbell Island record (Turney et al., 2018) (b and c); Macquarie Island and Neumayer Station (Levin et al., 2010).

*"In this context it might be necessary to recalculate the Southern Hemisphere atmospheric background of Δ14CO2, from which the anomalies of individual sites are calculated."*

Modelling studies such as (Graven et al., 2012) and (Levin et al., 2010) all show lower $\Delta^{14}C$ values over the Southern Ocean, that are attributed to the upwelling of old, $^{14}C$-depleted waters. The records from 41°S (Baring Head and Cape Grim) are thought not to be significantly influenced by the Southern Ocean signal (Levin et al., 2010). Therefore they are an appropriate anchor point.

Further, Baring Head/Cape Grim datasets from 41°S are the most complete and detailed records available in the Southern Hemisphere, making them more reliable as a reference than any other record.
* * ** * ** * *
*2. For the back-trajectories (Figure 4) the Southern Ozean is split in different zones according to the positions of the Antarctic Polar Front and other fronts as published in Orsi et al 1995. My understanding of positions of fronts, widely based on Freeman and Lovenduski (2016, doi:10.5194/essd-8-191-2016) is that the definition of these fronts might be different when based on newer studies. Thus, it might be necessary to revise the calculations of back-trajectories and if and how this might help to distinguish which water masses might influence via sea-air gas exchange the measured 14C data. This might also influence Figure 6.*

| Paper | Is data available? | What fronts do they map? |
|---|---|---|
| (Freeman & Lovenduski, 2016) | I have emailed Dr Freeman to see if the climatological data is publicly available. | PF |
| (Belkin & Gordon, 1996) | Don't think so. | STF, SAF, PF |
| (Moore et al., 1999) | Data not available, no supp. Info is found | PF |
| (Dong et al., 2006) | Data not available, no supp. Info is found | PF |

(Freeman and Lovenduski, 2016) makes their weekly polar front (PF) positions available, which can be averaged to yield the climatology. Rerunning our analytical scripts using the (Freeman and Lovenduski, 2016) versus (Orsi et al., 1995) PF produced Table 1 (A), the relative percentage of time that HYSPLIT back trajectory points spend in each Southern Ocean zone. The only zones affected by this analysis are the polar frontal zone (PFZ) and Antarctic Southern Zone (ASZ), as the PF divides them and all other positions remain from Orsi et al. (1995).

Table 1 (C) shows that for New Zealand sites, using the 2002-2014 (Freeman and Lovenduski, 2016) PF position means that *more* back-trajectory time is spent in the Antarctic Southern Zone and *less* in the Polar Frontal Zone, with larger impacts on southerly sites such as Campbell Island (swings of 5.6%) and lesser impacts on more northerly sites (down to 2.1% at Baring Head). This is driven by the (Freeman and Lovenduski, 2016) PF climatology being further north at 130°E and 50°E, compensating for other smaller shifts to the south (see Fig. 2). For Chilean sites, the impact is reversed, as PFZ is expanded rather than reduced due to the (Freeman and Lovenduski, 2016) PF shifting south around 90°W.

GLODAP mean Δ14C and nitrate concentrations are reanalysed to compare the PF positions (see Table 2). GLODAP data is temporally and spatially sparse. Thus, one additional point falling on either side of the PF can cause significant shifts between the two analyses.

Because our data spans the climatological range of both (Orsi et al., 1995) and (Freeman and Lovenduski, 2016), our results will reflect the mean of them both. The shifting polar front only affects calculated values for the PFZ and ASZ, and we will describe this in the manuscript accordingly. Figures and tables have been updated to reflect this change.

[Figure]

Fig 2. Overlay of (Orsi et al, 1995) Southern Ocean fronts, with polar front highlighted in blue, and (Freeman and Lovenduski, 2016) polar front highlighted in red.

| Analysis on March 11, using (Freeman and Lovenduski, 2016) climatological polar front position | | | | | | | | |
|---|---|---|---|---|---|---|---|---|
| | | | Freeman PF | | Orsi PF | | Difference | |
| Site | Latitude | Country | PF | ASZ | PF | ASZ | PF | ASZ |
| Puerto Navarino, Isla Navarino | -54.9 | Chile | 11.0 | 12.2 | 9.0 | 14.2 | 2.0 | -2.0 |
| Baja Rosales, Isla Navarino | -54.9 | Chile | 11.0 | 12.4 | 9.3 | 14.1 | 1.7 | -1.7 |
| Monte Tarn, Punta Arenas | -53.7 | Chile | 10.3 | 10.8 | 8.5 | 12.6 | 1.8 | -1.8 |
| Seno Skyring | -52.5 | Chile | 8.6 | 9.1 | 7.3 | 10.4 | 1.3 | -1.3 |
| Tortel River | -47.8 | Chile | 5.2 | 7.0 | 4.4 | 7.7 | 0.8 | -0.8 |
| Tortel Island | -47.8 | Chile | 5.5 | 7.0 | 4.7 | 7.8 | 0.8 | -0.8 |
| Bahia San Pedro | -40.9 | Chile | 5.3 | 5.5 | 4.7 | 6.1 | 0.6 | -0.6 |
| Campbell Island | -52.5 | NZ | 9.7 | 30.0 | 15.3 | 24.4 | -5.6 | 5.6 |
| Oreti Beach | -46.4 | NZ | 6.2 | 21.9 | 10.0 | 18.1 | -3.8 | 3.8 |
| Haast Beach | -43.9 | NZ | 4.6 | 13.7 | 7.2 | 11.1 | -2.6 | 2.6 |
| Eastbourne 2 | -41.3 | NZ | 4.1 | 14.1 | 6.2 | 11.9 | -2.1 | 2.1 |
| Eastbourne 1 | -41.3 | NZ | 4.1 | 14.1 | 6.2 | 11.9 | -2.1 | 2.1 |
| Baring Head | -41.1 | NZ | 4.1 | 11.8 | 6.2 | 9.7 | -2.1 | 2.1 |

Table 1. The percent of total time HYSPLIT back-trajectories spend in each Southern Ocean Zone using the (Freeman and Lovenduski 2016) polar front position, the (Orsi et al., 1995) polar front position and (C) the difference between the two.

| | Atlantic Ocean | | | | Pacific Ocean | | | | Indian Ocean | | | |
|---|---|---|---|---|---|---|---|---|---|---|---|---|
| Zone | Mean DIC $\Delta^{14}C$ | σ | Mean Nitrate | σ | Mean DIC $\Delta^{14}C$ | σ | Mean Nitrate | σ | Mean DIC $\Delta^{14}C$ | σ | Mean Nitrate | σ |
| **(Freeman and Lovenduski, 2016) polar front position** | | | | | | | | | | | | |
| *Only PFZ and ASZ are affected by changes in the PF position. | | | | | | | | | | | | |
| PFZ | -39.3 | 26.7 | 20.6 | 2.9 | -23.7 | 20.2 | 21.4 | 2.0 | -34.0 | 25.0 | 20.3 | 4.0 |
| ASZ | -82.4 | 17.6 | 22.2 | 2.3 | -74.5 | 23.3 | 26.1 | 2.3 | -96.1 | 23.5 | 26.8 | 2.9 |
| | | | | | | | | | | | | |
| **(Orsi et al., 1995) polar front position)** | | | | | | | | | | | | |
| PFZ | -28.1 | 19.7 | 20.1 | 2.7 | -22.0 | 20.1 | 21.8 | 2.5 | -29.1 | 28.0 | 19.1 | 4.2 |
| ASZ | -80.9 | 18.4 | 22.2 | 2.4 | -72.3 | 24.8 | 25.7 | 2.5 | -93.3 | 25.7 | 26.6 | 2.9 |
| | | | | | | | | | | | | |
| **Difference** | | | | | | | | | | | | |
| *Only PFZ and ASZ are affected by changes in the PF position. | | | | | | | | | | | | |
| PFZ | -11.2 | 6.9 | 0.5 | 0.2 | -1.7 | 0.1 | -0.4 | -0.6 | -4.9 | -3.0 | 1.1 | -0.1 |
| ASZ | -1.6 | -0.7 | 0.0 | -0.1 | -2.1 | -1.4 | 0.4 | -0.3 | -2.8 | -2.2 | 0.2 | -0.1 |

Table 2. Comparison of GLODAP mean Δ14C and nitrate concentrations using (Freeman and Lovenduski, 2016) vs (Orsi et al., 1995) PF position.

[Figure]

Fig 4. Updated manuscript Fig. 6 to include recalculated data and both polar fronts visible on the map.
* * *
* * *
* * *
*3. The overall motivation of the study, that these new 14C data might be used as proxies for Southern Ocean upwelling, which is already mentioned in the title, might need more support.*

We may consider changing the title since this is the first study on what would be a long-road toward tree-rings as upwelling proxies.

*My concern is based on the fact that not only the upwelling water mass in the Southern Ocean is depleted in 14C (old), but also the anthropogenic CO2 flux which operates in the opposite direction (atmosphere->ocean), since this is mainly based on 14C-free fossil fuel emissions. If you look at the atmospheric D14CO2 time series in Hua et al*

*2022 you see that the southern most data for the last decades have actually higher values than the northern ones. I believe this is understood by the fact that the 14C-free CO2 emission (14C Suess Effect)being mainly in the NH, but I cannot remember a citation to this idea. Here, it is now argued that the small scale differences between some sides in the SH can be used to pin down the upwelling flux of old waters. I think it is necessary to discuss that what is concluded here on smaller scale (upwelling of old 14C-depleted waters reduces atm D14CO2 in some places around the Southern Ocean) is opposite (and maybe counterintuitive) to the more global picture (14C Suess effect larger in N than in S).*

At the global scale, our $^{14}CO_2$ observations and modelling studies all demonstrate that there is a double-dip in $\Delta^{14}C$ (i.e. high $\Delta^{14}C$ values in the Arctic, a marked decrease in the Northern Hemisphere midlatitudes (attributed to the Suess Effect/fossil fuel $CO_2$ emissions), an increase again in the tropics (due to lesser fossil fuel emissions, and heterotrophic respiration from the biosphere), then another decrease at high Southern latitudes associated with Southern Ocean upwelling, and finally another increase associated with the decreasing influence of Southern Ocean upwelling AND the more common stratospheric incursions near the South Pole ((Graven et al., 2012; Levin et al., 2010; Randerson et al., 2002; J. Turnbull et al., 2009; J. C. Turnbull et al., 2017)).

Intuition suggests that the influence of fossil fuel emissions/Suess Effect will be minimal over the Southern Ocean, since there are so few emissions in this region. One could argue that the transport of Northern Hemisphere fossil fuel emissions could create a gradient in the Southern Hemisphere, but this is not supported by the modelling studies, previous observations, and our study. We do not observe a significant latitudinal gradient across the lower latitudes of the Southern Hemisphere (say 20-45°S), but do observe this gradient at the higher southern latitudes. This can only be explained by the Southern Ocean.

*Minors:*

*- Line 40: Citation to „Peter Landschutzer, 2015" is weird (first name?) and missing in the reference list.*

This has been fixed.

*- The DOI to 10.5281/zenodo.14532802 (Data Availability) does not work.*

The Zotero link has been fixed, and the DOI is now 10.5281/zenodo.14532801
* * *
**RC2**: 'Comment on egusphere-2024-4107', Anonymous Referee #2, 27 Jan 2025  reply

*This study lays the groundwork for the use of Southern Hemisphere tree-rings for studying changes in Southern Ocean upwelling and air-sea $CO_2$ flux. The authors demonstrate that trees from Chile and New Zealand from the 1980s to present accurately record atmospheric 14C (with a few exceptions) with a latitudinal gradient observed for New Zealand sites.  Using HYSPLIT software they model back-trajectories for the air masses reaching the site and show that the New Zealand gradient is driven by Southern Ocean outgassing and atmospheric transport.  The methods used are robust and the manuscript is generally well-written.  Due to the importance of the Southern Ocean in global climate, this study will be critical for future work on constraining temporal changes in upwelling and air-sea gas exchange.*

*Minor comments: (strikethrough indicates completion/comment is resolved)*

*Line 30: Insert space after (Talley, 2013)* Done

*Line 33: Possibly insert 'content' after 'carbon' just to be clear that this doesn't refer to 14C (although probably fine).* Done

*Table 1: 'The final column references numbered locations on Figure 2a/b maps.'  Note: The final column with location number is missing.* Done

*Line 163: 'using a FFT filter cut-off of 667'  What are the units?*
The value for fc is specified in 'number of days'. Figure 4 shows the response function with cutoff values of 80 days for the short term filter and 667 days for the long term filter.
https://gml.noaa.gov/ccgg/mbl/crvfit/crvfit.html

*Line 168: 'Data will be included in Supplementary Material'  Is this yet to be done or will it be at 10.5281/zenodo.14532802?*

I've simply removed this line because it may cause confusion. I will let readers find the Data Availability statement rather than bring it up piecemeal throughout the paper.

*Figure 2:  Perhaps change the label 'CCGCRV Trend Reference' to 'CCGCRV SHB Trend' for clarity*

It may be clearer to simply remove the legend. An extra line was added in Figure caption to say the line has been smoothed.

*Line 209: 'The data was filtered for existing radiocarbon measurements ('G2c14')' Please insert 'ocean' in front of radiocarbon.*

Following text was edited:
The GLODAP Merged and Adjusted Data Product v2.2023 is used to add context to the discussion of our results (Key et al., 2004; Lauvset et al., 2023; Olsen et al., 2016). The data was filtered for existing ocean dissolved inorganic carbon radiocarbon

measurements ('G2c14') south of 5°S,  shallower than 100m, and post-1980 to match our tree-ring temporal span.

*Line 213: 'on bottle latitude/longitudes'  Some readers may not be familiar with the use of bottles for ocean water collection- perhaps replace with 'sample collection latitude/longitudes'*

Fixed

*Line 309: '(Prend et al., 2022) discovered a deep maximum mixed layer depth surrounding Campbell Island and westward in the ASZ (Prend 2022 Fig. 1c).'  The sentence should start with 'Prend et al. (2022) discovered' and presumably end with (Prend et al. 2022, Fig. 1c)*

Fixed

*Line 313: 'Low ΔΔ14CO2 at Campbell Island may be linked air originating'. Insert 'to' after 'linked.*

Fixed

*Line 326: 'local effects such as nearby obduction from deep mixed layer' Doesn't obduction refer to crustal movements?*

Prend et al. (2022) define obduction as follows:
"We find that the interbasin variation in surface ocean pCO2 is primarily driven by regional variability in wintertime entrainment of carbon-rich deep water from the permanent pycnocline ("obduction")." (Prend et al., 2022)

*Line 329: 'This suggests that the influence of non-local fossil fuel and biospheric signals are small in the Southern Hemisphere.'  Does this apply to the entire Southern Hemisphere?*

Yes. This response in in alignment with previous observational and modelling studies that show the latitudinal gradients of 14C from pole-to-pole suggests that fossil fuel emissions and biosphere signals are small in the Southern Hemisphere. 14C increases in the tropics due to lower fossil fuel emissions and biospheric signals, but decrease in the Southern Hemisphere due to Southern Ocean upwelling before increasing again toward the pole due to stratospheric production of 14C ((Graven et al., 2012; Levin et al., 2010; Randerson et al., 2002; J. Turnbull et al., 2009; J. C. Turnbull et al., 2017)).

*The citations need to be carefully checked as many in the text are missing from the reference section or incomplete.  I've found the following problems with citations but may have missed some:* Fixed

Line 110: Zondervan et al., 2016.  This is given as 2015 in references

Line 200: Warner 2018 is not in references.  Should this be Millissa & Warner 2018?

Line 490: 'Norris, M. W. (2015). Reconstruction of historic fossil CO2 emissions using radiocarbon measurements from tree rings'  This reference is incomplete.

Citations not in reference section:

Line 52: Turnbull et al. 2009

Line 58: Rend et al. 2022 (typo, should have been Prend et al., 2022).

Line 108: Baisden et al. 2013

Line 109 Turnbull et al. 2015

Line 110: Stuiver & Polach 1977

Line 161: Thoning et al. 1989

Line 182: Suess 1955

**RC3**: 'Comment on egusphere-2024-4107', Anonymous Referee #3, 08 Feb 2025  reply

This study aims to reconstruct atmospheric $\Delta^{14}C$ variations near the Southern Ocean to infer past changes in Southern Ocean upwelling rates. Given the crucial role of Southern Ocean overturning circulation in the climate system, this is a highly relevant and important objective. The authors present a latitudinal $\Delta^{14}C$ gradient derived from tree-ring records in Chile and New Zealand, spanning the past several decades. By applying backtracking approaches, they distinguish contributions from different Southern Ocean regions and assess the reliability of this proxy. This approach appears promising, and ground-truthing studies are essential to validate proxy-based reconstructions. As such, the study deserves publication, though some clarifications are necessary before acceptance.

The authors report a meridional $\Delta^{14}C$ gradient in the New Zealand records, with increasing $^{14}C$ depletion toward the south, which they attribute to the upwelling of $^{14}C$-depleted waters from the Southern Ocean. In contrast, no significant gradient is observed in the Chilean records. They suggest this discrepancy arises because tree-ring records from Chile are less influenced by the degassing of polar surface waters. Although that it is supported by atmospheric back-trajectory modelling, a more thorough description on the systematics of the Southern Ocean $^{14}C$-$CO_2$ exchange is somehow lacking in the text. This exchange is complicated and may explain some of the difference between New Zealand and Chile $\Delta^{14}C$.

We will add more descriptive text in the manuscript regarding this point.
"Southern Ocean carbon flux is complex and variable, with different sectors acting as sources and sinks for anthropogenic and natural CO2, respectively (Gruber et al., 2019). The Southern Ocean as a whole acts as a sink for anthropogenic CO2, with uptake concentrated in the PFZ (Mikaloff Fletcher et al. 2006, DeVries 2014; Gruber et al., 2019). In the case of natural CO2, higher latitudes, specifically those of the ASZ and PFZ, dominate outgassing while lower latitudes such as in the STZ, dominate uptake (Gruber et al., 2019). Some zones outgas seasonally, while the ASZ outgasses nearly year-round (Gray et al., 2018). Uptake and outgassing are also zonally asymmetric. The Indian and Pacific sectors jointly dominate Southern Ocean outgassing, with the Pacific specifically dominating in the PFZ (Prend et al., 2022). These complexities play key roles in facilitating a signal of one-way flux of low Δ14C from the surface ocean toward distant trees."

*The error bars in Fig. 3 are large, raising questions about the significance of the New Zealand trend. Please specify in both the figure caption and the main text what these error bars represent and how they were calculated (e.g., 1 or 2 standard deviations). Reporting the p-value would also help assess the statistical significance of this trend. Additionally, it would be useful to display individual observations—perhaps using smaller, empty, or slightly shaded symbols—to allow readers to better evaluate the robustness of the trends.*

These error bars represent 1 standard deviation. The p-value for the trendline of New Zealand site *means* is 0.014, while that of Chile is 0.395 (calculated using the SciPy *linregress* module). The New Zealand p-value being <0.05 indicates the trend is statistically significant, while the Chilean trend is not. We will add text to the paper to indicate this all more clearly. I will confer with co-authors as to whether the figure below with semi-transparent symbols should appear in the main text or supplementary information (it may seem too busy for the main manuscript); however, if the latter, I will highlight for readers to seek in the Supplementary Material.

[Figure]

*The Cape Grim and Baring Head records are considered as representative of the Southern Hemisphere record and serve as the baseline for the Southern Hemisphere atmospheric background. It would be interesting to provide a more detailed discussion on their representativeness by comparing them with other global and Southern Hemisphere records, especially that the authors emphasize several times that that their study significantly expands Δ14C records for the Southern Hemisphere. Additionally, since Cape Grim and Baring Head are influenced by Southern Ocean air masses, could lower-latitude Southern Hemisphere records provide a more representative baseline?*

As (Hua et al., 2021) is a compilation, I can directly compare my work with the individual publications *within* the compilation. I will explore this moving north to south, beginning with the latitude of Wellington (41°S).

**Latitude 41S: Wellington vs Cape Grim Australia**

Something we did not adequately make clear in the first version of this manuscript is that tree-cores from Baring Head and Eastbourne from which $\Delta\Delta^{14}CO_2$ is calculated *are* tree cores from Turnbull et al. (2017), which Hua et al., (2021) incorporates into their manuscript. We will make this clear in the subsequent version. In Turnbull et al. (2017), the authors discuss how the Wellington and Cape Grim records are "generally consistent with one another" despite a period of time where instrument difficulties led to anomalous values at the Rafter Radiocarbon Laboratory (RRL). In Turnbull et al. (2017) and in this work, that period of time is removed and replaced with the University of Heidelberg Cape Grim record for the Southern Hemisphere Background (Levin et al., 2010).

We also present a direct site-site intercomparison in which Cape Grim and Baring Head are directly compared and with $\Delta^{14}CO$ measured in the same laboratory to test interlaboratory offsets. Although the temporal range is short, we also find no difference between the site (Fig. 1(c).). These lines of evidence provide confidence that there is no difference between the Wellington and Cape Grim records.

**Campbell Island 52.5°S.**

*Atmospheric records* exist on Campbell Island but extend from 1970 to 1977 (Manning et al., 1990), ruling out a direct comparison with our Campbell Island *tree-ring record*.

Another Campbell Island tree-ring record exists (Turney et al., 2018) which includes records from the same *Sitka spruce* and *Dracophyllum*. The *Sitka spruce* record only extends to 1967, disallowing a direct comparison with our records from the same tree.

If the public access *Dracophyllum* records are analyzed according to our workflow, we find there is a difference between the *Dracophyllum* record (Turney et al., 2018) and the Campbell Island record presented in this work (see Fig. R1(c)). Independent t-test of $\Delta\Delta^{14}C$ yield a p-value of $<<0.05$, indicating that the two datasets are robustly different. Visual inspection shows the (Turney et al., 2018) records seem to deviate from our record around 1990 and 2007. Even if these periods of deviation are removed from the t-test, the data are still robustly different. Mean $\Delta\Delta^{14}C$ (from our workflow) for (Turney et al., 2018) data is $0\pm2.6‰$ (no difference from Southern Hemisphere background) while our record shows Campbell Island is $-3.1\pm3.3‰$ from background.

What would lead to different results from tree-ring measurements on the same island? Fossil fuel contamination would push $\Delta^{14}C$ values lower, while biospheric $CO_2$ would increase $\Delta^{14}C$. Since the island is uninhabited, there is little chance of fossil fuel contamination decreasing our results; however, sustained high westerly winds also make biospheric contamination in (Turney et al., 2018) similarly unlikely. One additional source of uncertainty is that *Dracophyllum* and *Sitka spruce* have different tree-ring

widths and potentially different growth periods which may lead to differences in records from differing species, even from the same island.

One key aspect of the work we present is that Campbell Island is compared to other locations which were sampled and processed the same way, by the same people, and measured in the same AMS system. This means that Campbell Island record is best viewed *relative* to sites from the same work.

**Macquarie Island and Neumayer Station**

We will add in the next manuscript version a direct comparison between our Campbell Island record and the (Levin et al., 2010) Macquarie Island record (see Fig. R1(d,e)). This comparison is instructive because they are in close proximity. Independent t-tests confirm that the Campbell Island tree-rings and the Macquarie Island atmospheric measurements are not statistically different (p = 0.3). Macquarie Island record (if put through our analytical workflow) yields -2.6±3.2‰, similar to Campbell Island $\Delta\Delta^{14}$C.

This agreement between Macquarie Island and Campbell Island demonstrates that our result is realistic, and justifies our use of a longer Campbell Island record.

Neumayer Station similarly is found to be no different from Campbell Island (p=0.5) with a mean $\Delta\Delta^{14}$C of -2.7±5.2‰.

[Figure]

**Fig R1.** A comparison of Campbell Island tree-ring $\Delta^{14}$C with other Southern Hemisphere sites including another Campbell Island record (Turney et al., 2018) (b and

c); Macquarie Island and Neumayer Station (Levin et al., 2010).

**On the representativeness of Baring Head and Cape Grim as a baseline**

Modelling studies such as (Graven et al., 2012) and (Levin et al., 2010) all show lower $\Delta^{14}C$ values over the Southern Ocean, that are attributed to the upwelling of old, $^{14}C$-depleted waters. The records from 41°S (Baring Head and Cape Grim) are thought not to be significantly influenced by the Southern Ocean signal (Levin et al., 2010). Therefore they are an appropriate anchor point.

Further, Baring Head/Cape Grim datasets from 41°S are the most complete and detailed records available in the Southern Hemisphere, making them more reliable as a reference than any other record.

---

## Author Response (AR2)

1. Range of data is sometimes 1980-2016, sometime 1980-2017, sometimes 1980 until today (abstract, section 2.4, conclusions). Please homogenize.
**Writing has been homogenized to say 1980-2017 in all 3 sections.**

2. Levin et al. 2010 is given for reference of Heidelberg data from Cape Grim, e.g Fig 1. However, there are data after 2010 contained in Fig 1, which cannot be part of Levin et al. 2010. So, either add another paper for them or a data reference where the data are found.

**Additional citation added:**
**Levin, I., Hammer, S., 2022. Supplementary data to Levin et al. (2022), Continuous measurements of 14C in atmospheric CO2 at Cape Grim, 1987-2016. https://doi.org/10.18160/8F31-EQDJ citation has been added to cover Cape Grim data post 2010.**

Additional notes

**Additional slight changes to text have been made for clarity on lines 263-265 and 278-280 (Line numbers refer to document with track changes).**

"In Turnbull et al., (2017), the authors publish Baring Head and Eastbourne tree-ring Δ14C measurements, and provide a rigorous comparison to show that the Wellington Δ14CO2 record agrees well with atmospheric measurements from Cape Grim, Australia (Levin et al., 2010, 2022). An additional two-year site-site intercomparison provides confidence that Cape Grim and Baring Head are not different (Fig 1. (c))."

"However, an important consideration is that the records and results we show were all sampled, processed, and measured identically, meaning results from Campbell Island are best viewed relative to other sites from this work. Additionally, low values at Campbell Island can be validated by matching the Macquarie Island record (see below), where interlaboratory offsets between the two institutions have been investigated and none found (see Baring Head versus Cape Grim above and Hammer et al., (2017))."